# Development of Chitosan Green Composites Reinforced with Hemp Fibers: Study of Mechanical and Barrier Properties for Packaging Application

**DOI:** 10.3390/molecules28114488

**Published:** 2023-06-01

**Authors:** Rim Gheribi, Yassine Taleb, Louise Perrin, Cesar Segovia, Nicolas Brosse, Stephane Desobry

**Affiliations:** 1Laboratoire d’Ingénierie des Biomolécules, Université de Lorraine, ENSAIA, 2 Avenue de la Forêt de Haye, BP 20163, 54505 Vandœuvre-lès-Nancy, CEDEX, France; rim.gheribi@univ-lorraine.fr (R.G.); yassine.taleb6@etu.univ-lorraine.fr (Y.T.); louise.perrin@univ-lorraine.fr (L.P.); 2Centre d’Essais TEchnique LORrain (CETELOR), 27 rue Philippe Seguin, BP 21042, 88051 Épinal, CEDEX 9, France; cesar.segovia@univ-lorraine.fr; 3Laboratoire d’Etudes et de Recherche sur le Matériau Bois, Faculté des Sciences et Technologies, Université de Lorraine, Boulevard des Aiguillettes, BP 70239, 54506 Vandœuvre-lès-Nancy, CEDEX, France; nicolas.brosse@univ-lorraine.fr

**Keywords:** mechanical properties, barrier properties, packaging materials, circular economy

## Abstract

The use of bioresourced packaging materials is an interesting solution for ecological issues. This work aimed to develop novel chitosan-based packaging materials reinforced with hemp fibers (HF). For this purpose, chitosan (CH) films were filled with 15%, 30%, and 50% (*w*/*w*) of two kinds of HF: Untreated fibers cut to 1 mm (UHF) and steam exploded fibers (SEHF). The effect of HF addition and HF treatments on chitosan composites was studied in terms of mechanical properties (tensile strength (TS), elongation at break (EB), and Young’s modulus (YM)), barrier properties (water vapor (WVP) and oxygen permeabilities), and thermal properties (glass transition (T_g_) and melting temperatures (T_m_)). The addition of HF, whether untreated or steam exploded, increased the TS of chitosan composites by 34–65%. WVP was significantly reduced by the addition of HF but no significant change was observed for O_2_ barrier property, which was in the range between 0.44 and 0.68 cm^3^·mm/m^2^·d. T_m_ of the composite films increased from 133 °C for CH films to 171 °C for films filled with 15% SEHF. However, no significant modification was observed for T_g_ (105–107 °C). The present study showed that the developed biocomposites had improved properties, mainly the mechanical resistance. Their use in food packaging will help industrials the move toward a sustainable development and circular economy.

## 1. Introduction

Nowadays, concerns for environmental safety, ecosystem preservation, and climate change are within the priorities of researchers and industrials. In this context, the use of natural and bioresourced materials, also called biomaterials, is an interesting and promising answer for ecological concerns [1]. These materials are respectful to the environment and human health, they are relatively light in order that their carbon impact could be reduced, and they are biodegradable and/or methanizable by some bacteria species and under controlled conditions [2]. However, in a practical way, this solution remains challenging and difficult to set up. The reasons behind the difficulties of using biomaterials are their cost and technical limits, especially for industrial applications. Biomaterials generally present poor mechanical and barrier properties. Moreover, their low resistance to high temperatures limits industrial application [1].

Several bioresourced polymers were studied and their ability to develop packaging biomaterials was investigated. Among these biopolymers, we found starch, chitosan, alginate, pectin, gelatin, caseinate, and cellulose derivatives [3,4]. Chitosan is one of the most interesting and promising biopolymers for the development of new generation materials. After cellulose, this bioresourced polysaccharide derived by deacetylation from chitin is the second most abundant biopolymer in nature [5]. Chitin is mainly extracted from crustaceans’ exoskeletons, insects’ exoskeletons, or fungi cell walls. When its degree of deacetylation (DDA) is above 50%, chitin turns into chitosan which is a copolymer made of two structural units: N-acetyl-D-glucosamine and D-glucosamine. Some of its numerous characteristics include biodegradability, biocompatibility, and non-toxicity. Moreover, chitosan is interesting for physicochemical properties (cationic character), antimicrobial and antioxidant capacities, and its ability to be easily modified chemically. Today, chitosan is used in different forms whether as foam, gels, films, or granules and in different fields, namely, cosmetics, medicine, and wastewater treatment [5,6].

For food packaging applications, chitosan was used through different processing methods, such as solvent casting, dip- and spray-coating, layer-by-layer preparation, and extrusion [6]. These chitosan-based materials present good thermal stability and promising physicochemical properties. However, their mechanical and barrier properties are poor and not suitable for food packaging applications. These findings were confirmed by several previous studies, which reported that chitosan films are fragile, highly hygroscopic, and sensitive to environmental variations [7,8,9,10].

One of the newly used methods for enhancing the mechanical properties of films is the combination of biopolymers with natural fibers. In fact, natural fibers are lignocellulosic materials with high aspect ratio, and are commonly used for their low cost, durability, good mechanical properties, low weight, and biodegradability [6]. Therefore, natural fibers bring mechanical strength to the polymeric material without altering its ecofriendly attributes. Composites reinforced with natural fibers have a lower environmental impact thanks to their reduced carbon and fuel consumption. Hemp fibers (*Cannabis sativa*) are very abundant plant fibers with a world production of 214 × 10^3^ tons per year and France is one of the main producers of this natural biomass [6,11]. Hemp fibers are basically composed of cellulose (70–74%), hemicellulose (15–20%), and lignin (3.5–5.7%). Even though their mechanical properties depend on various factors, such as plant growing conditions and physical, chemical, and morphological properties, hemp fibers present an interesting tensile strength in the range between 550 and 1110 MPa [11]. Consequently, hemp fibers are considered to be one of the strongest and stiffest natural fibers. Mechanical performance is strongly related to interfacial adhesion between the polymeric matrix and fibers, to the dispersion of fibers in the main matrix, and to the fiber properties, such as dimension, orientation, and surface modification [11]. Hemp fibers have already been used for the reinforcement of several materials, such as polypropylene, polylactic acid, and polybenzoxazine [12,13,14], in particular, to replace glass fibers; however, they were rarely used in combination with biobased polymers, such as polysaccharides. Several chemical and mechanical treatments have been tested to improve their dispersion and their interfacial adhesion with the polymeric matrix [11]. Steam explosion is one of the mechanical methods and has been described as a method for degumming and producing fine, homogeneous hemp fibers [15,16]. It was shown that this process degraded the middle lamella of the cells by hydrolysis of the pectins and hemicelluloses that it contained.

Chitosan films reinforced with glass, sisal, ramie, or bamboo fibers were studied in the literature [17,18,19]; however, the incorporation of HF with different mechanical treatments was never investigated. In the present paper, we aimed to study the development of chitosan-based films reinforced with hemp fibers for their potential application as food packaging materials. Untreated and steam exploded hemp fibers were used for this purpose and their effect on mechanical, barrier, and thermal properties was investigated. This work represents a feasibility study in order to have a proof of concept of our hemp fiber-reinforced biomaterials. 

## 2. Results and Discussion

### 2.1. General Description and Microstructure

Chitosan–hemp fibers composite films (Figure 1A,C) produced in the current study were translucid with a rough touch on the upper side and a smooth touch on the inner side (in contact with the petri dish). Their general appearance was homogeneous and repeatable; however, we could observe that films loaded with UHF were more homogeneous and those filled with SEHF were smoother. No cracks were observed which confirms the cohesiveness and homogeneity of the plasticized composite films. Microstructural properties of pure chitosan films have already been studied in literature [20]. Chitosan films have a compact and homogeneous structure and the surface is smooth and flat. This homogeneous microstructure is generally related to the strong intra- and inter-molecular hydrogen bonds occurring between chitosan functional groups on the one hand, and chitosan, water, and glycerol molecules on the other hand [20].

Once the fibers are added, the composite microstructure was completely different as shown in Figure 1B,D, which represents films with 50% UHF and 50% SEHF, respectively. Micrographs of films filled with 50% of fibers are represented and are representative of the disposition and orientation of fibers in the film structure. The mentioned micrographs showed that, for both treatments, the fibers were randomly dispersed within the film structure, without any precise orientation. Film surface morphology was continuous and homogeneous for both kinds of hemp fibers. SEHF were dispersed more uniformly in the film matrix and covered the entire film surface. In fact, during steam explosion, fibers are blown apart and split into microfibrils due to the process of high pressure and the breakdown of pectin, hemicellulose, and lignin [21,22]. In contrast, the mechanical treatment only modified the length of hemp fibers, but not their chemical or structural properties. The dispersion of UHF in film matrix was less effective and less homogeneous than SEHF.

### 2.2. Chemical Characterization by Infrared Spectroscopy

To better understand the contribution of hemp fibers to the molecular structure of chitosan composites, fibers were analyzed by FTIR spectroscopy (Figure 2a). Both UHF and SEHF showed similar bands which are specific to plant fibers mostly composed of cellulose, hemicellulose, lignin, and pectin. The broad band observed around 3270–3410 cm^−1^ is assigned to hydroxyl groups of the lignocellulosic material. Peaks observed between 2850 and 2910 cm^−1^ are related to C–H groups of the fibers. C=C stretching bands related to carboxylic or unsaturated acids were detected at 1630 cm^−1^ for UHF and at 1730 and 1639 cm^−1^ for SEHF. This was the main difference shown between UHF and SEHF, which is related to hemicelluloses hydrolysis and delignification during the steam explosion process, as reported by Nader et al. [23]. C–H and C–H_2_ bending peaks related to cellulose and lignin were observed at 1200–1430 cm^−1^ for both fibers. Two peaks were detected at 1051 and 1026 cm^−1^ and were attributed respectively to the C–O–C vibration and C–O stretching present in the cellulose structure. A last band was observed at 890 cm^−1^ and was assigned to the C–H vibration of cellulose. All the mentioned peaks specific to lignocellulosic plant fibers were reported in previous works and agreed with our FTIR characterization [24,25].

The developed films were analyzed and the related FTIR spectra are presented in Figure 2b. The obtained graphs showed that all composite films had almost the same peaks with some minor variations. CH-based films showed characteristic bands of this branched polysaccharide, mainly amide groups (amide I at 1645 cm^−1^, amide II at 1552 cm^−1^, and amide III at 1311 cm^−1^), C–H stretching vibration at 2916 and 2850 cm^−1^, C–O symmetric stretching vibration at 1406 cm^−1^, C–O–C bridge antisymmetric stretching at 1151 cm^−1^, and C–O stretching at 1020 cm^−1^. These bands were also identified for chitosan composites by Moalla et al. [20] and Teixeira-Costa et al. [26]. The addition of hemp fibers did not bring any new peaks, which confirms the absence of covalent bonds, but all peaks were shifted to lower wavenumbers which indicates the presence of hydrogen bindings between chemical groups of chitosan polymer and those of hemp fibers (O–H stretching vibration detected mainly at 3250–3350 cm^−1^). The presence of shifts was detected at 3338, 3356, and 3350 for films with UHF loaded at 15, 30, and 50%, respectively, and observed at 3352, 3355, and 3357 for films with 15, 30, and 50% SEHF, respectively. Moreover, water and glycerol contribute to the creation of these low-energy linkages.

### 2.3. Mechanical Properties

Figure 3 summarizes the tensile strength (TS), elongation at break (EB), and Young’s modulus (YM) of the different composites. The neat chitosan films, developed at 1.5% with a chitosan with a viscosity of 12 mPa·s, present poor tensile strength with values around 32 MPa. These values are comparable to those obtained for films developed with chitosan with different molecular weight, and thus with different viscosities (20–40 mPa·s) [27]. EB did not exceed 3% which was a low value, but is similar to the results obtained by Liu et al. [27]. The addition of 30% UHF did not significantly (*p* < 0.05) affect TS of chitosan-based composites in comparison to the control film (CH). At this ratio, the fibers could not be well-dispersed in the film-forming suspension and surface attraction between the two compounds decreased [9]. However, the addition of hemp fibers, whether untreated or steam exploded, significantly increased the tensile strength for the other tested ratios (15% and 50% UHF, 15%, 30%, and 50% SEHF). The best result was obtained for 50% (*w*/*w*) UHF and SEHF, where tensile strength values exceeded 50 MPa (Figure 3). The highest TS values were obtained for 50% of hemp fibers (for both UHF and SEHF). The addition of hemp fibers improved the tensile strength of chitosan films by 34–65%, which means that this method is an efficient, green, and cost-effective way to develop chitosan materials with enhanced mechanical properties. However, it is worth mentioning that the steam explosion treatment improves the compatibility between HF and chitosan, thanks to the high aspect ratio of the steam exploded fibers and their large surface areas, which gives an abundance of hydroxyl groups on the surface [22,28]. Therefore, steam explosion promotes the good and uniform dispersion of individual fibers within the polymeric matrix, which was confirmed by SEM micrographs, and leads to a better mechanical resistance of the composite materials. As UHF gives less homogeneous film structures, the effect of their addition at three different ratios on the mechanical properties of composites could not be relevant.

For EB and YM, no significant increase was observed for all the developed films. However, the obtained results agreed with previous studies reporting that chitosan composites had low EB and YM [26,27]. For the packaging field, the development of CH–HF-based materials with good mechanical resistance could fit well with the requirements of the industrial application, where good mechanical resistance is mandatory.

### 2.4. Barrier Properties

Barrier properties related to water vapor and oxygen are important parameters for food packaging materials due to the potential diffusion of these two gases through the material from the internal or external environment. Chitosan control films, plasticized with 10% (*w*/*w*) glycerol, showed a WVP of 7.5 g·mm/m^2^·d·kPa (equal to 86.8 × 10^−12^ g·cm/cm^2^·s·Pa), which is comparable to the values obtained by Liu et al. [27] (~2 × 10^−12^ g·cm/cm^2^·s·Pa for films based on chitosan with different molecular weights). The addition of hemp fibers reduced significantly the WVP of the studied composites, in comparison with the control film (Table 1). The decrease in WVP was observed for all composite films. However, the lowest WVP values were observed for the films where SEHF were added, especially at 50% (*w*/*w*). Steam explosion treatment removes hemicellulose from hemp fibers, which makes it less hydrophilic, and thus decreases its affinity to water molecules. Therefore, the more important the fiber content, the better the water vapor barrier property. It was reported that steam explosion increases the porosity, and thus the tortuosity of fibers, which makes water diffusion more difficult [16]. Therefore, water molecules did not diffuse directly through the film and passed through the different pores of treated HF, which slowed down its crossing and improved its barrier properties.

For OP, the addition of HF increased film permeability in comparison with control CH films. Incorporation of treated or untreated HF created micropores in the polymeric matrix, which made it more porous, and thus more permeable to gas, such as O_2_, CO_2_, and N_2_. However, all OP results are less than 1 cm^3^·mm/m^2^·d·kPa, which is considered to be a very good barrier property to oxygen, as confirmed by previous studies reporting that chitosan- and polysaccharide-based materials in general, exhibit good gas barrier properties. This could be explained by the well-ordered hydrogen-bonded network in the matrix of films [9].

### 2.5. X-ray Diffraction

X-ray diffractograms of the CH + HF films were used for the measurement of the degree of crystallinity of the films. As presented in Figure 4, the X-ray diffractogram of neat HF displayed diffractogram peaks at 2θ = 15.5, 22.5, and 34.0, associated with a semi-crystalline structure in the amorphous matrix of fibers. The average crystallinity of the HF was 61.8%. Considering chitosan film, the significant peaks are located at 2θ = 23.5, 34.1. The large peak was due to the wide range of molecular mass of chitosan molecules which was crystallized (75.8% of crystallinity ratio). A major type of crystal appeared (2 theta = 23°). 

### 2.6. Thermal Properties

In Table 2, glass transition temperatures (T_g_) ranged from 105 to 110 °C, with no significant differences. The presence of a single T_g_ for each composite confirms the good miscibility of the different components of film matrix [10]. However, the addition of SEHF significantly increased the melting temperatures (T_m_) of the composites, which increased from 134 °C for CH control films to 171, 167, and 142 °C for films filled with 15%, 30%, and 50% of SEHF (*w*/*w*), respectively. The shift of T_m_ to higher temperatures by the addition of SEHF could be related to the higher thermal resistance and stability of fibers subjected to steam explosion treatment due to the elimination of hemicellulose (decomposition at 260 °C) and the higher content of cellulose (decomposition at 360 °C) [11]. Moreover, the increased crystallinity of cellulose in the case of steam exploded fibers improves their thermal stability [22]. The high content of free water in film-forming solutions containing 15% of fiber compared to solutions with 30% and 50% SEHF favored crystallization during film formation. Films with 15% of SEHF are more crystallized than those with 30 and 50% SEHF. Crystallization had no effect on T_g_, but increased the melting temperature (T_m15_ > T_m30_ > T_m15_). T_m_ decreased with the increase in SEHF concentration. 

## 3. Material and Methods

### 3.1. Material

Chitosan was purchased from Matexcel (New York, NY, USA) with a degree of deacetylation of 60%, viscosity of 12 mPa·s, and purity of >97%. Glycerol (as a plasticizer) and acetic acid (as a solvent) were purchased from Sigma Aldrich (St. Louis, MO, USA).

Industrial hemp fibers (*Cannabis sativa* L.) were cultivated and harvested by La Chanvrière Company (Troyes, France).

### 3.2. Methods

#### 3.2.1. Fiber Extraction and Treatments

Hemp fibers (HF) used for the development of chitosan-based composites were subjected to two different extraction methods: Mechanical extraction (industrial process) without any chemical or enzymatic treatment (UHF) and steam explosion (SEHF). 

For mechanical treatment, hemp fibers were first opened with the LAROCHE opening machine using the first drum at 800 rpm and low-feeding speed. They were then introduced in the BONINO carding machine at a speed of 15 m/min via a loading system to create a web. The web was fed through the coiler of the carding machine to achieve a sliver of oriented hemp fibers, at a weight of 20 g/m. This sliver was then taken to the Pierret textile cutter, where the fibers were cut to the desired length.

Steam explosion was performed using the method of Chadni et al. [29]. Raw hemp fibers, previously soaked in distilled water for 16 h, were placed into a 2 L reactor with heat jacket and automatic control for steam pressure and sampler residence time. Fibers were steam exploded at 200 °C and 15 bars for 4 min. The steam-saturated biomass was released into a discharge tank. The obtained fibers were later filtered, washed, and dried at 20 °C for 72 h. 

#### 3.2.2. Films Development

Films were developed by the casting method according to Liu et al. [27]. Film-forming solutions were prepared by dissolving chitosan (1.5%, *w*/*w*) in acetic acid solvent (2%, *v*/*v*) at 70 °C under continuous stirring. Glycerol was then added as plasticizer at 10% (*w*/*w*, based on chitosan dry weight). For composite films, HF whether cut to 1 mm or steam exploded, were added to the CH film-forming solution at 15, 30, or 50% (*w*/*w*, according to CH dry weight) and the suspensions were stirred for 1 h. Film-forming solutions were cast into plastic petri dishes (14 cm in diameter), oven-dried for 1 h at 50 °C, and then left to dry for 48 h in a conditioned room at 25 °C and 40% RH. All films were stored at 50% RH and 25 °C prior to mechanical and barrier characterization tests.

#### 3.2.3. Films Characterization

##### Fourier Transform Infrared Spectroscopy (FTIR)

IR spectroscopy was performed on a Tensor 27 mid-FTIR spectrometer (Bruker, Germany) with a Deuterated Triglycine Sulfate (DTGS) detector and a diamond Attenuated Total Reflectance (ATR) module. FTIR spectra were recorded between 4000 and 400 cm^−1^ with 64 scans and a resolution of 4 cm^−1^. Dried film samples of 1 × 1 cm were used and the analysis was performed in triplicate.

##### Scanning Electron Microscopy (SEM)

Surface morphology of films was studied by SEM using a JEOL JSM 6010LA high-resolution scanning electron microscope (Tokyo, Japan). Prior to the analysis, film samples were metallized with gold and the analysis was conducted with an accelerating voltage of 5 or 10 kV.

##### Thickness

The film thickness was determined using a Mitutoyo micrometer (Tokyo, Japan) and the analysis was performed 10 times on different positions of each film.

##### Water Vapor Permeability (WVP) 

This characterization was performed based on the gravimetric method using ASTM standard method E96/E96 M [30] with modifications (T = 25 °C; ΔRH = 75%). Briefly, film disks (with a surface of 27 cm^2^) were placed on hermetic permeation cells filled with 50 g of silica gel. The cells were placed in a chamber containing a saturated solution of sodium chloride to obtain a relative humidity of 75% within the chamber at 25 °C. Weight gain (water adsorption in silica gel) was followed-up for 2 weeks and tests were conducted in triplicate. Water vapor transmission rate (WVPR) was determined by linear regression based on the slope of the cell weight change per square meter of film vs. time (R^2^ > 0.99) during the steady state.

The WVP (in g·mm/m^2^·d·kPa) was calculated as shown in Equation (1):WVP = WVTR × e/Δp(1)
where WVTR is the slope (g/m^2^·d), e is the film thickness (mm), and Δp is the gradient of partial water pressure (Δp = 2.38 kPa).

##### Oxygen Barrier Properties

Oxygen permeability was investigated using an oxygen analyzer (Systech Illinois Instrument, Thame, UK) according to the ASTM D3985-05 method [31] on disk films with a surface of 78.5 cm^2^. The oxygen transmission rate (OTR) was measured by the analyzer software. Moreover, the oxygen permeability was calculated at steady state according to Equation (2) and expressed in cm^3^/m^2^·d·kPa. Tests were performed in triplicate at 23 °C and 50% RH.
OP = OTR × e/Δp (2)
where OTR is the oxygen transmission rate, e is the thickness of the film, and Δp is the partial pressure difference across the film (kPa).

##### Mechanical Properties

Mechanical characterization was studied in terms of tensile strength (TS, MPa), elongation at break (E, %), and Young’s modulus (YM, MPa). This characterization was performed using an universal tensile machine (Shimadzu AGS-X, Kyoto, Japan), based on the ASTM D882 standard method [32]. Films were cut into rectangular samples of 15 mm × 100 mm and the test was performed with a head speed of 10 mm/min using a double clap separated by 50 mm. Five replicates were tested for each film.

##### Differential Scanning Calorimetry (DSC) 

Thermal properties of chitosan-based composites were studied using a DSC (DSC 2500 Discovery Series from TA Instruments, Hüllhorst, Germany). Film samples (around 40 mg) were sealed in aluminum pans and tests were conducted in triplicate under nitrogen atmosphere. Heating/cooling/heating cycles were run from 0 °C to 180 °C with heating and cooling rates of 10 °C/min.

##### X-ray Diffraction (XRD)

Crystalline properties of neat chitosan films, untreated hemp, and steam exploded fibers were studied by means of an X-ray diffractometer (BRUKER D8 Advance) using the following conditions: Cu Kα radiation (wavelength of 1.54056 Å) from the anode operating at 40 kV and 40 mA, with a graphite monochromator, an angular range of 2θ = 5–120° with a step width of 0.02°/scan. The location and the intensity of the obtained peaks were used for the determination of crystalline ratio (ratio between crystal peaks and amorphous changes in the base line).

##### Statistical Analysis

All experimental data were subjected to one-way ANOVA analysis with 95% significance level using Origin software (Origin Lab, version 2019, Northampton, MA, USA). Duncan’s test was performed to detect the significant differences (<0.05).

## 4. Conclusions

The effect of different concentrations and treatments of HF on the properties of chitosan films was studied. Mechanical resistance and WVP of composite films were significantly improved by HF addition. TS reached more than 50 MPa for films with 50% of UHF and SEHF. WVP results were particularly interesting for SEHF-charged films. The lowest WVP values were measured for CH films with 50% of SEHF. Similarly, films with SEHF showed higher melting temperatures, which made them suitable for material processing in the packaging industry. This indicated that chitosan and SEHF had good miscibility and compatibility thanks to the fact that steam explosion is a physical method that changes the structural and surface properties of the fiber. The good miscibility of CH and SEHF was supported by SEM, FTIR, XRD, and DSC analyses. 

The current study showed that when adding treated or untreated HF, the functional properties of chitosan films were improved, which makes them promising for food packaging applications as films or catering containers. Steam explosion seems to be a good mechanical treatment to improve microstructural surface, to favor crystallinity, and promote interactions between chitosan and SEHF. Incorporating SEHF at 50% based on the chitosan polymer could be particularly interesting to improve functional properties, reduce material processing costs, and respect sustainable development goals. 

## Figures and Tables

**Figure 1 molecules-28-04488-f001:**
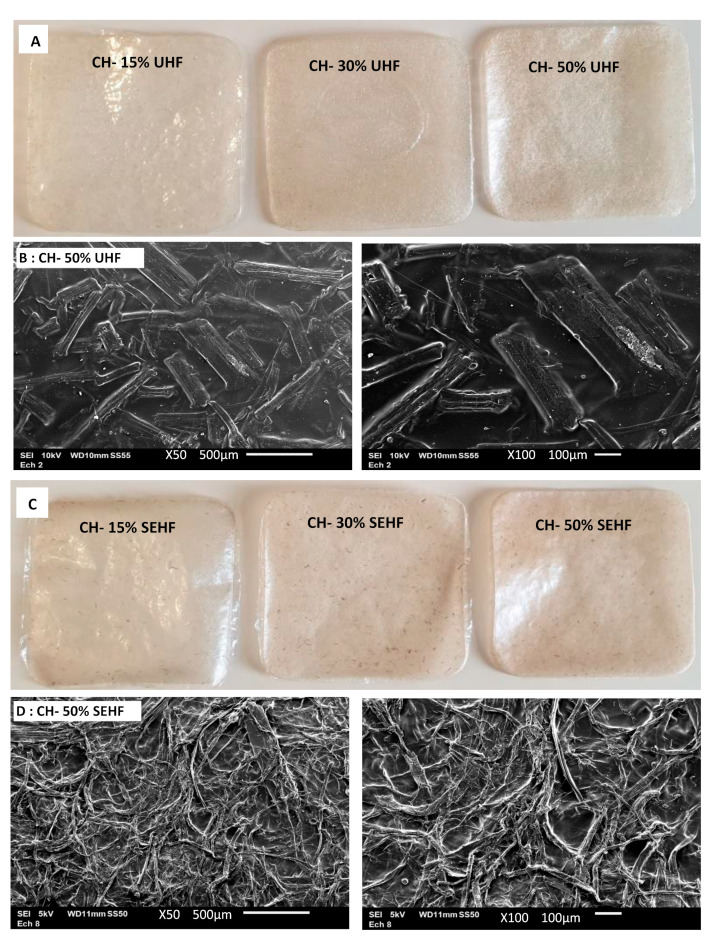
General appearance of chitosan films reinforced with untreated fibers (UHF, (**A**)) and steam exploded fibers (SEHF, (**C**)). Micrographs of films with 50% HF show the fibers at film surface (**B**,**D**) at two different magnifications (×50 on the left and ×100 on the right).

**Figure 2 molecules-28-04488-f002:**
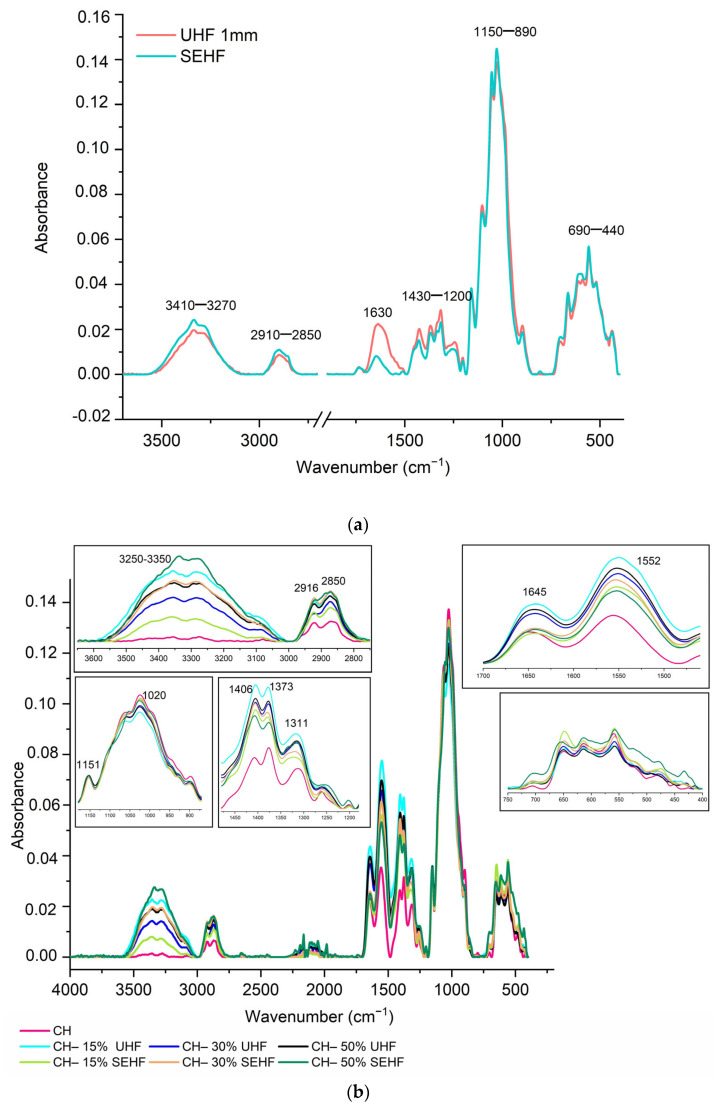
FTIR spectra of untreated and steam exploded hemp fibers (**a**) and the different composites of chitosan–hemp fibers (**b**).

**Figure 3 molecules-28-04488-f003:**
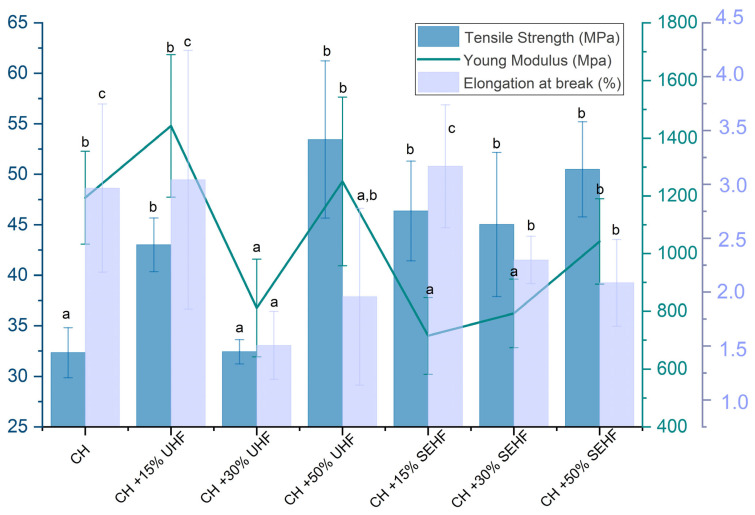
Mechanical properties of the different composites of chitosan–hemp fibers; values with different letters are significantly different (*p* < 0.05).

**Figure 4 molecules-28-04488-f004:**
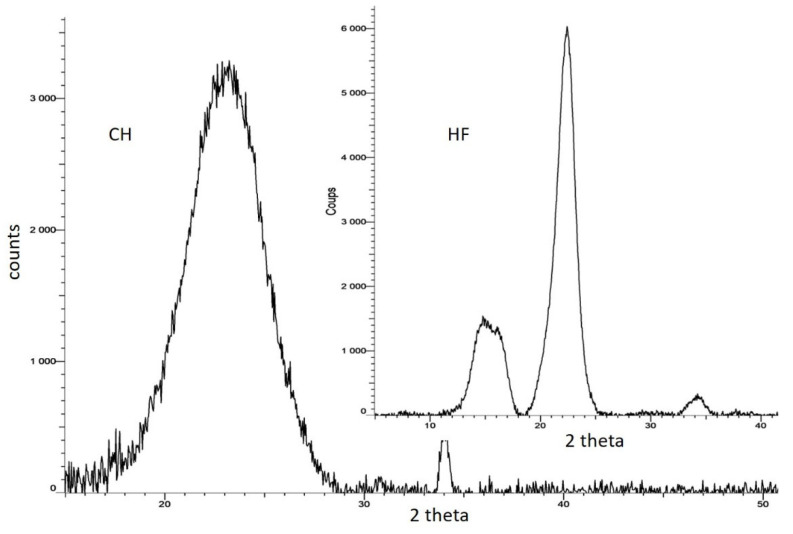
X-ray diffractograms of chitosan (CH) and hemp fibers (HF).

**Table 1 molecules-28-04488-t001:** Thickness, water vapor, and oxygen permeabilities of the different chitosan–hemp fibers composites.

	Thickness (µm)	WVP (g·mm/m^2^·d·kPa)	OP (cm^3^·mm/m^2^·d·kPa)
CH	167.5 ± 3.5 ^b^	7.5 ± 0.2 ^a^	0.44 ± 0.17 ^a^
CH-15% UHF 1 mm	146.8 ± 6.0 ^a^	5.4 ± 0.2 ^c^	0.49 ± 0.04 ^b^
CH-30% UHF 1 mm	227.2 ± 13.2 ^c^	6.5 ± 0.1 ^b^	0.79 ± 0.11 ^d^
CH-50% UHF 1 mm	271.8 ± 21.0 ^d^	6.6 ± 0.1 ^b^	0.68 ± 0.18 ^c^
CH-15% SEHF	225.0 ± 21.4 ^c^	5.6 ± 0.004 ^c^	0.67 ± 0.17 ^c^
CH-30% SEHF	238.8 ± 23.7 ^d^	5.3 ± 0.2 ^c^	0.64 ± 0.22 ^c^
CH-50% SEHF	235.1 ± 22.8 ^c^	3.5 ± 0.1 ^d^	0.65 ± 0.21 ^c^

CH: Chitosan control (1.5% *w*/*w*), UHF: Untreated hemp fibers cut to 1 mm, SEHF: Steam exploded hemp fibers, WVP: Water vapor permeability, OP: Oxygen permeability; values are means of n replicates ± standard deviation (*n* = 10 for thickness; *n* = 3 for WVP and OP); values with different letters in the same column are significantly different (*p* < 0.05).

**Table 2 molecules-28-04488-t002:** Glass transition and melting temperatures of the different composites.

	Glass Transition Temperature T_g_ (°C)	Melting Temperature T_m_ (°C)
CH	105 ± 0.4 ^a^	133 ± 0.1 ^a^
CH-15% UHF 1 mm	105 ± 0.4 ^a^	136 ± 0.5 ^a^
CH-30% UHF 1 mm	105 ± 0.2 ^a^	138 ± 0.1 ^a^
CH-50% UHF 1 mm	105 ± 0.4 ^a^	133 ± 0.1 ^a^
CH-15% SEHF	106 ± 0.7 ^a^	171 ± 0.1 ^c^
CH-30% SEHF	110 ± 3.1 ^b^	167 ± 0.1 ^c^
CH-50% SEHF	105 ± 0.9 ^a^	141 ± 1.2 ^b^

CH: Chitosan control (1.5% *w*/*w*), UHF: Untreated hemp fibers cut to 1 mm, SEHF: Steam exploded hemp fibers; values with different letters in the same column are significantly different (*p* < 0.05).

## Data Availability

Data sharing is not applicable to this article.

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
