# Peer review of "Development of Chitosan Green Composites Reinforced with Hemp Fibers: Study of Mechanical and Barrier Properties for Packaging Application"

_molecules, 2023, doi:10.3390/molecules28114488_

Round 1

Reviewer 1 Report

In the present manuscript, the authors reported and demonstrated that the addition of hemp fibers (HF) and steam explored hemp fibers (SEHF) improved mechanical, thermal and barrier properties of chitosan. Several characterization techniques were employed to investigate the influence of added fibers. Some interesting results were obtained, and I recommend its publication in Molecules after carefully major revision.

My detailed comments are given below:

1. The author mentioned that the interfacial adhesion between modified fibers and matrix directly affects the mechanical properties in Line70, however, there is no relevant characterization and discussion in the section of results and discussion.

2. Figure 1 does not clearly indicate which sample is modified by HF or SEHF.

3. Line 217, the addition of 30% UHF did not affect TS of composites, because fibers could not be dispersed in the film forming dispersion at this ration. While the authors successively discussed that the addition of 15% and 50% UHF significantly improves the tensile strength of composites. The dispersion of different addition ratios and corresponding influence on the structure of composites should be discussed.

4. Adding HF fibers significantly improved the TS of chitosan films by 34-65%, and related mechanism should be investigated. And SEM should be employed to study the compatibility of different components.

5. Line 251, the authors mentioned that incorporation of fibers created pores, however there are no relevant characterizations. SEM and BET might be proper techniques.

6. Line 276, the addition of SEHF increased the Tm of composites, and the more fibers added, the less Tm increased. Why does this phenomenon appear? And the influence of addition rations different from that of Question 3.

Moderate editing of English language (e.g. Line19 weather might be whether?)

Author Response

Reviewer 1

My detailed comments are given below:

  1. The author mentioned that the interfacial adhesion between modified fibers and matrix directly affects the mechanical properties in Line70, however, there is no relevant characterization and discussion in the section of results and discussion.

Dear reviewer, thank you for your relevant comments; SEM analysis was added and used for a better comprehension of chitosan- hemp fibers interaction and adhesion. We used this new analysis for a better interpretation of mechanical results (results and discussion section, line 191).

  1. Figure 1 does not clearly indicate which sample is modified by HF or SEHF.

Figure 1 was modified.

  1. Line 217, the addition of 30% UHF did not affect TS of composites, because fibers could not be dispersed in the film forming dispersion at this ration. While the authors successively discussed that the addition of 15% and 50% UHF significantly improves the tensile strength of composites. The dispersion of different addition ratios and corresponding influence on the structure of composites should be discussed.

We totally agree with you; for this reason, we analyzed the surface morphology by SEM and we added some modifications on results section according to the obtained results.

  1. Adding HF fibers significantly improved the TS of chitosan films by 34-65%, and related mechanism should be investigated. And SEM should be employed to study the compatibility of different components.

As mentioned before, we actually added SEM characterization for a better understanding of the interaction and compatibility of the different components (L. 190-205).

  1. Line 251, the authors mentioned that incorporation of fibers created pores, however there are no relevant characterizations. SEM and BET might be proper techniques.

Same response as the previous comment

  1. Line 276, the addition of SEHF increased the Tm of composites, and the more fibers added, the less Tm increased. Why does this phenomenon appear? And the influence of addition rations different from that of Question 3.

This phenomenon was discussed in section 3.6 and modifications were added in L354-358.

Reviewer 2 Report

 Dear Editor,

Thank you for giving me opportunity to express my opinion over the manuscript molecules-2365379.

 I have carefully read the manuscript entitled “Development of Chitosan green composites reinforced with Hemp Fibers: Study of mechanical and barrier properties for packaging application” and analyzed its potential for publication in the MDPI journal Molecules (ISSN 1420-3049).

In my opinion, the manuscript has a potential but it is way too modest to pass as scientific article.

 I suggest rejecting the manuscript.

I base my opinion on the following arguments:

1. Introduction does not present a reasonable overview of the current state of research on the topics discussed in the work.

2. The work isn’t supported by a reasonable number of cited works (13 cited articles )

3. I have failed to find reasonably formulated research goals

4. The work lacks a scientific approach to the researched issues

5. Conclusions are very weak and ill-considered

6. The work is mechanical research, complete lack of considerations at the molecular level, which seems to me inconsistent with the scope of the journal

Author Response

Dear reviewer,

Thank you very much for your revision and for your argumentation.

In fact, this paper is part of a project and the main goal of the project was a proof of concept to design Chitosan- Hemp fibers composite for potential industrial application as cosmetic and food packaging; this is the reason why this study seems to be scientifically modest and lack of research goals. We actually brought major modifications to this paper according to the comments of all reviewers and add new results (SEM) to enrich the paper. We hope that you have a better opinion about the modified version of our study. References and conclusions were also modified.

Thanks again,

The authors.

Reviewer 3 Report

The work deals with the  development of green nanocomposites based on chitosan and hemp fibers, untreated and treated by steam explosion. Chitosan was effectively reinforced through HP addition, and characterization was carried out (Structural, thermal, barrier and mechanical properties). The work has novelty and results are very interesting for readers of “Molecules” journal and researchers working on reinforced food packaging. Nonetheless, it is highly important improved discussions and make major revisions before publishing. The revisions are indicated below:

Line 17-18: Abbreviations of elongation at break, Young modulus and oxygen permeability are missed. Also, author have the option of avoiding abbreviation in the abstract, and indicate them in the next sections.

Line 19: Does “untreated of steam exploded” have a problem of typing?

Line 24: It is better to specify with properties were improved.

Line 35-36: Cite general references on the poor properties of biomaterials.

Lines 51-55: Please, include references on processing techniques and production of chitosan film with poor properties.

Line 68. Cite reference about TS of HF ranging between 550 and 1110 MPa.

Line 78: ¿Which were the two kinds of hemp fibers? ¿Which properties of the development materials you aimed to measure?

Separate the number and the unit with a space. Revise all manuscript. For example, in line 94, “2 L”.

Line 96: ¿Which was the steam pressure used in the steam explosion?

Revise the absence of articles like “the”, “a”… in the text (English grammar)

Line 107: ¿Do you have information on dimensions of the films? Please, indicate if that’s possible.

Line 124: Change uppercase to lowercase “Water” to “water”.  Unit related to the thickness film is missed in the indicated units.

Line 125,130, 136…: Change “KPa” to “kPa”

Line 130: Change “film Thickness” to “film thickness”

Line 130,141: ¿Which was the value of ΔP?

Line 134: ¿What “ref.” means?

Line 137: Indicate %RH

Line 136: Please, revise units. Is the unit of thickness film in relation to equation 2 missed? Is it correct “m2.J.kPa”?

Line 139: Number of the equation is missed.

Line 149: Indicate number of ASTM normative.

Line 149: ¿Do the ASTM standard indicate that 5 replicates is enough to have a good result? or ¿Were there limitations of sample quantity?

Line 298. A space between “the” and “different” is missed

In Figure 2, the legend indicates seven samples (fig b) and two samples (Fig.a), however, six spectra are distinguished in the graph area of Fig. 2b and two spectra in Fig. 2a.  Color of lines CH and CH 50% UHF are very similar. Please, used other color to a better visualization. Please, clarify legends.

Line 194: Please, rephrase the sentence to a better understanding

Line 262-263: Please, could you give the value of the displacement of the peaks? Considering that in the Figure is difficult to observe them.

In Figure 3, in X-axis, ¿Is the name of the control sample “CH”? Ch 1.5%? Please, verify and correct along manuscript, figures and tables.

Line 212: ¿Were films at 1.5% of HF made? This 1.5% is confused here and in the X-axis of the figure 3.

In the mechanical properties, ¿Why is the cause of the differences in the trends of the tensile strength between composites with UHF and SEHF? Specifically, at 30% of HF. Something in the experimental preparation or some further explanation?. Discussion about changes in YM and E with the different HF and concentration should be given. ¿Is there a effect of particle agglomeration or the chemical structure/adhesion?

Line 219-220: It would be interesting to include a microscopy analysis to observe the dispersion of the fibers in the sample.

Line 241: Please, indicate the value reported by Liu et al. (2020) for WVP that is comparable with the results to a better understanding. Comparisons with the same units.

Line 242: Use WVP after the first mention of this abbreviation

Line 247-249: Please, the explanation on how porosity of the fiber increases the tortuosity in the film should be improved. It is not clear. ¿Do you refers to a porosity of the fiber or the films? A greater porosity of the fibers could be allowing a better adhesion with CH and strengthen the interphase reducing the possibility of water vapor path?

Table 1. Correct KPa a kPa. Unify code or name of the CH control. CH or CH (1.5% w/w), 100% CH, Ch 1.5%. Add 1.5% is confusing. The foot of the table shows the mean of OTR but the table reports OP. Please, correct it.

Line 254: Cite reference about the good barrier property value OP that is claimed.

Line 264-266: Sentence seems to be incomplete or should be rephrased.

Line 269: The sentence “On major type of crystal appeared for a 75.8% of crystallinity ratio”  is difficult to understand. Please, explain more clearly.

Include in the methodology about XRD, a brief procedure of crystallinity calculation and clarify which samples were analyzed.

Line 263: XRD analysis was carried out for the films with CH and HP?. In the methodology, is indicated that only neat CH film was analyzed by XRD. Discussion of XRD should be improved to clarify the assignation of the reported crystallinity values.

Lines 270-271: This explanation should be supported relating crystallinity of the films (CH+HP) with barrier properties results.

Table 2: Based ob the statistical differences and trends, it would be better report temperature values without decimals. It is not necessary due to the trends are clearly observed.

Discussions of thermal properties section can be improved. ¿Is there relation between the different interactions between UHF and SEHF with CH (displacements of bands in FTIR) with differences in the influence on Tm? ¿Which kind of interactions with CH considering that SEHF was less hydrophilic?. ¿Differences in crystallinity of the films were considered to explain the results?

Line 286: Change “amounts” for “concentrations”.

Line 240: “better thermal resistance” should be erased, and it is more appropriate refer specifically to Tm, which was the measured parameter.  ¿Does TGA also carried out?

Line 297: ¿What does the claim “to reduce material processing cost” means? ¿How this conclusion is related with the obtained results?

In summary, the conclusion should be improved avoiding general conclusions, and indicating which composites are preferably recommended, with HF or SEHF.

Minor editing of english language is required

Author Response

see attached file (word)

Round 2

Reviewer 1 Report

The revised draft has answered all my questions. Therefore, I suggest to accept this paper.

Author Response

The reviewer 1 validated the paper. We nevetheless enriched the article with answers to comments from reviewers 2 and 3.

Reviewer 2 Report

I cannot find the substantial improvent of the manuscript in terms of scientific soundness of the paper

Author Response

We added references to enrich the paper intrroduction and discussion. We rewrite large parts of the article (see green ink parts). We hope that these improvements will reach your expectation.

Reviewer 3 Report

The authors made a great job and improve the manuscript which topic and research work is very interesting to be published. Nonetheless, I think there are some minor revisions which need to be carried out before publishing:

1. In Eqs. 1 and 2 change "slope" for "WVTR" and "OTR", respectively.

2. Line 145: I think "J" is a mistake in the units. Should it is time units?

3. Figure 1: The SEM micropgraphs are very interesting. Thanks for include them. Please, indicate in Figure Caption the concentration of HP for the SEM micrograph shown in each case. A better explanation to related Figs. A and C with Figs. B and D should be given. The explanation (Lines 182-188) is only focused in the structure of control CH reported by SEM in the literature, and not in the included figures.

4. Lines 275-277: Please, if it is possible, indicate the used conditions (%HR and T) for the WVP of the cited reference, to better support the differences with your values. 

5. Section 2.2.3: Add space between paragraphs of each characterization to a better readability of the text.

6. Section 3.5: Please, remove decimals of the temperatures values in the text according to the values in the table. Furthermore, I think it is important to report crystallinity of the films containing fibers, calculated for DSC, to better support differences in Tm or other finding. On the contrary, better explain the claims in lines 323-326. It is confusing because a reduction of the Tm could be related with a reduction of the crystallinity but as it was not measured, maybe hydrogen bonding interactions reduced mobility and chain ordering at the higher concentrations?. Please, clarify the discussions.

Author Response

Thanks for your very attentive reading of the paper. We made all changes and improvements you proposed. The paper is now highly improved, thanks to your comments.